# Individual and interpersonal factors influencing child marriage: A qualitative content analysis study

Asma pourtaheri[1], Mehr Sadat Mahdizadeh[2,3], Hadi Tehrani[2,3], Jamshid Jamali[3,4], Nooshin Peyman[3,5] *

1 Ph.D. Candidate of Health Education& Health Promotion, Student Research Committee, Mashhad University of Medical Sciences, Mashhad, Iran, 2 Department of Health Educationand Health Promotion, School of Health, Mashhad University of Medical Sciences, Mashhad, Iran, 3 Social Determinants of Health Research Center, Mashhad University of Medical Sciences, Mashhad, Iran, 4 Department of Epidemiology and Biostatistics, School of Health, Mashhad University of Medical Sciences, Mashhad, Iran, 5 Professor of Health Educationand Health Promotion, Department of Health Educationand Health Promotion, School of Health, Mashhad University of Medical Sciences, Mashhad, Iran

* peymann@mums.ac.ir

**Data Availability Statement:** All relevant data are within the paper and its Supporting information files.

## Abstract

### Background

Child marriage is one of the public health challenges that has caused increasing concerns in the health and development system. Therefore, this research was conducted to identify individual and interpersonal factors influencing child marriage from the perspectives of stakeholders and informants.

### Method

This qualitative study was conducted using content analysis and an inductive approach from 2023 to 2024 in Bam city, Kerman, Iran. Thirty-six stakeholders (girls who have been married for 15 years, parents, husbands, and informants) were purposively selected. Data were collected through semi-structured interviews and analyzed using the Graneheim and Lundman approach, with the assistance of MAXQDA software.

### Results

After analyzing of the data, individual and interpersonal themes were emerged. The first theme of "individual factors" consisted of biological, psychological, and demographic category with four sub-categories including insufficient cognitive and inferential development, physiological and anatomical features, facing stressful factors in life, and demographic characteristics. The second theme of "interpersonal factors" consisted of family structure with four sub-categories including traditional parenting methods, family values, family breakup, Inefficiency of management and problem-solving in the family, and weak social capital in the family. The category of Ineffective interactions and social support also encompass two subcategories: Peer pressure and reference groups, and inappropriate care and support relationship between teachers and students.

**Funding:** The author(s) received no specific funding for this work.

**Competing interests:** The authors have declared that no competing interests exist.

**Abbreviations:** CM, Child Marriage; HDI, Human Development Index; SDG, Sustainable Development Goals; SRQR, Standards for Reporting Qualitative Research; STD, Sexually Transmitted Diseases; UNICEF, United Nations Children's Fund.

## Conclusion

The results showed that individual and interpersonal factors are effective on children's marriage. Some individual factors have a biological origin, indicating that increasing girls' awareness of marriage, pregnancy, individual rights, and life skills is one solution that can help reduce early marriage. On an interpersonal level, fostering positive relationships within the family, school, and society, and strengthening the support network can play a crucial protective role for children.

## Introduction

Child Marriage (CM) refers to the marriage of boys and girls before the age of 18, which has attracted the attention of policymakers and researchers for a long time [1]. Although the term applies to both boys and girls, girls are more commonly affected by this practice [2]. According to UNICEF, the rate of marriage under the age of 18 is 21% for young women (aged 20–24) and 3% for young men [3]. It is a harmful practice that violates children's rights, which remains widespread despite the decline in the global prevalence of CM over the past decade. Annually, 12 million girls worldwide are married before the age of 18 [4]. Therefore, today we have approximately 650 million girls who are married under the age of 18. The highest levels of CM are observed in sub-Saharan Africa (38%), South Asia (30%), and Latin America (25%) [5].

Considerable literature has shown that CM has several harmful consequences for young women. These include loss of educational opportunities [6], intimate partner violence [7], sexually transmitted diseases (STDs) [8, 9], mental health issues [10], complications during pregnancy and childbirth [11], perinatal mortality, and infant mortality [12]. The concern about the harmful personal and social consequences of CM has prompted the international community to address the issue of ending all forms of violence against women and CM in the 2030 Sustainable Development Goals (SDG) [13]. This goal aims to enhance maternal health, decrease child mortality, and advance gender equality. It is estimated that if CM is not curbed and prevented, over 150 million girls under the age of 18 will be married by 2030 [14].

Despite the increase in the average age of marriage among women in Iran [15], UNICEF reports that 17 percent of Iranian women marry before the age of 18 [16]. In Iran and around the world, numerous studies have been conducted in the field of CM. Mirzaee et al examined the viewpoints of stakeholders and found that factors such as the preservation of virginity and religious beliefs, prevention of social stigma, and legislation were the driving forces of CM in North Khorasan, Iran [17]. By examining the experiences of women who were married in childhood, Bozorgi et al showed that poverty, traditional barriers, escaping from the home environment, and satisfying the sexual needs of girls were among the effective factors in CM in Tehran [18]. Kohno's et al study revealed that health risk behavior, family poverty, early marriage as destiny, and family disharmony are the driving factors of CM in Sarawak, Malaysia [19]. Çelik et al introduced four factors of family coercion, love, poverty, and losses suffered during the war as factors affecting the early marriage of Syrian immigrant women [20]. In examining the health promotion approach, we find that individual and interpersonal factors play a crucial role in shaping behavior. Models such as social-cognitive theory and social network theory are based on this premise. However, to date, no study has specifically addressed the individual and interpersonal factors influencing child marriage. Also, this study focuses on the marriage of children under 15 years of age, a critical transitional period in their lives.

During this time, children move beyond the family unit and become immersed in a network of communication through school, their neighborhoods, and media, all of which can significantly influence their decisions. This underscores the necessity for further research in this area. Therefore, the current study aims to investigate the individual and interpersonal factors that affect CM.

## Materials and methods

The ontological position adopted in this study was critical realism. This perspective suggests that reality can be understood through the examination of the human mind and socially constructed meanings [21]. This approach was chosen for this study to uncover the individual and interpersonal factors that shape the context of CM. We followed the Standards of Qualitative Research Reporting (SRQR) guidelines [22] (S1 Table). This study was conducted to answer the following question.

a. What are the individual factors leading to CM?

b. What are the interpersonal factors leading to CM?

### Setting and context

This study was conducted in Bam City, Kerman, Iran. Kerman province in Iran is facing challenges in terms of the Human Development Index (HDI) (ranking 15th out of 31 provinces) [23]. Marriage among girls under 18 is prevalent in southern cities like Bam. The economy of the region relies on agriculture and a designated special economic zone that houses automobile and component factories. The population is predominantly Persian with a Baloch minority in terms of ethnicity, and a Shiite majority with a Sunni minority in terms of religion. In 2002, a devastating magnitude 6.8 earthquake claimed the lives of tens of thousands of people, resulting in significant changes in the economic, social, and structural landscape.

### Participants

This study utilized a qualitative approach of content analysis to elucidate individual and interpersonal factors influencing CM. Bam City has 12 health centers. After consulting with the University of Medical Sciences and the local government, the areas with the highest prevalence of child marriage in both the city and surrounding villages were identified. Sampling was purposefully conducted among girls who were married before the age of 15, as well as mothers, fathers, husbands, and informants from April 2023 to January 2024. It should be noted that in Iran, according to the law, girls can get married at the age of 13 with the permission of their parents and with the expediency of the court. In this study, we set the marriage age below 15 years as the criterion because, despite the legality of marriage under the age of 15, few people support it. To obtain the richest data, we ensured maximum diversity in terms of geographic location, age at marriage, and current age for eligible girls. Additionally, interviews were conducted with other stakeholders, including parents, spouses, and teachers associated with the students (S2 Table).

### Inclusion and exclusion criteria

The entry criteria were different for each group of participants. Eligible criteria for girls' entry included being married under the age of 15 and having been married for at least 5 years after the 2002 earthquake, being of Iranian nationality, residing in Bam city, and providing consent

to participate in the study. We specifically chose the 5-year mark after the earthquake to ensure that any lingering social effects of the earthquake on girls' marriages had dissipated. The informants included in the study were individuals involved in CM, with at least 5 years of work experience, who agreed to participate in the research. The inclusion criteria for parents and spouses included giving consent to participate in the study. Individuals who did not provide consent were excluded from the study in each group.

## Ethical considerations

This article is a part of the doctoral thesis on Health Education and Health Promotion, which has been approved by ID IR.MUMS.FHMPM.REC.1401.105 in Mashhad University of Medical Sciences. Before starting the interview, permission to record the audio was obtained, and the participants signed the informed consent form. We assured the participants that the information would remain confidential with the researcher and would be deleted after data extraction.

## Data collection

Semi-structured interviews and face-to-face methods were used to collect data. The interviews were conducted by a local person familiar with the cultural and social conditions of the region (first author). The interviews were conducted after prior coordination with the participants and obtaining the necessary permissions. Participants were asked about individual and interpersonal factors that lead to CM. Three sets of questions were designed for each group. For example, eligible girls were asked how did you decide to get married? Parents were asked how did you consent to your daughter's marriage? Spouses were asked how did you decide to marry a girl under 15 years old? Teachers were asked what do you think about the marriage of students? The duration of each interview varied from 45 to 60 minutes depending on environmental factors. After conducting 33 interviews, we discovered duplicate codes and did not discover any new concepts from the interviews. To ensure data saturation, we conducted three additional interviews. A total of 36 people participated in the study. Each interview lasted between 45 and 60 minutes, depending on environmental factors.

## Reliability

The researcher conformed to the Lincoln and Guba criteria for the validity, confirmability, portability and reliability by employing the following strategies: dedicating sufficient time to gather information, excluding personal opinions and biases, purposeful sampling with maximum diversity, conducting interviews until data saturation was reached. The implementation of the interview immediately after conducting it, detailed rewriting of the interview, extensive involvement in data and code extraction, providing a detailed step-by-step description of the research process, and consulting with professors were also part of the methodology [24].

## Data analysis

The data analysis was conducted using the approach developed by Graneheim and Lundman. This method involved the following five steps: 1) Transcribing each interview immediately after it was conducted, 2) Gaining a general understanding of the interview content through repeated readings, 3) Identifying semantic units and establishing basic codes, 4) Classifying primary codes to create more comprehensive categories, and 5) Determining the main topics and categories [25]. Initially, the interview text was read multiple times to gain a general understanding of the content. The coding process commenced by reading the text and

generating open codes. Codes were then grouped based on semantic and conceptual similarities to create more comprehensive categories. This analytical process continued until categories and themes were established. To ensure accuracy, an independent researcher was engaged to review the codes. In instances of unresolved disagreements between the primary investigator and the independent researcher, a third investigator was consulted to make a final decision. Microsoft Word software was utilized for transcription, and MAXQDA software was employed for analysis.

## Results

This study was conducted through semi-structured interviews with 36 eligible girls, parents, spouses, and informants. The participants included 15(41.66%) girls, 3 (8.33%) fathers, 6 (16.66%) mothers, 4 (11.11%) Husband, 5 (13.88%) teachers, and 3(8.33%) counselors. More details are shown in Table 1.

A total of 726 codes were extracted. After merging the codes, 82 final codes were obtained. After analyzing of the data, individual and interpersonal themes were emerged. The first theme of "individual factors" consisted of biological, psychological, and demographic category

**Table 1. Demographic characteristics of study participants.**

| Variables | Girls | Family Member | Informant |
|---|---|---|---|
| | Frequency (%) | Frequency (%) | Frequency (%) |
| **Age**, mean (SD) | 21.80 (4.76) | 43.62(13.45) | 45(6.28) |
| **Gender** | | | |
| Male | 0 | 7(53.84) | 0 |
| Female | 15(100) | 6(46.15) | 11(100) |
| **Education** | | | |
| Illiterate | 0 | 1(7.69) | 0 |
| student | 3(20) | 0(46.15) | 0 |
| High school | 5(33.3) | 5(38.46) | 0 |
| Diploma and postgraduate diploma | 6(40) | 5(38.46) | 0 |
| Expert and above | 1(6.6) | 2(15.38) | 11(100) |
| **Job** | | | |
| Employee | 0 | 4(30.76) | 11(100) |
| Homemaker | 13(86.6) | 4(30.76) | 0 |
| Farmer | 0 | 1(7.69) | 0 |
| Other | 2(13.3) | 4(30.76) | 0 |
| **Ethnicity** | | | |
| Persian (Fars) | 12(80) | 11(84.61) | 11100) |
| Baluch | 3(20) | 2(15.38) | 0 |
| **Religion** | | | |
| Shia | 12(80) | 11(84.61) | 11(100) |
| Sunni | 3(20) | 2(15.38) | 0 |
| **Residence** | | | |
| Urban | 8(53.3) | 6(46.15) | 11(100) |
| Rural | 7(46.6) | 7(53.84) | 0 |
| **Economic** | | | |
| Good | 5(33.3) | 8(61.53) | 11(100) |
| Medium | 7(46.6) | 3 (23.07) | 0 |
| Weak | 3(20) | 2(15.38) | 0 |

with four sub-categories including insufficient cognitive and inferential development, physiological and anatomical features, facing stressful factors in life, and demographic characteristics. The second theme of "interpersonal factors" consisted of family structure with four sub-categories including traditional Parenting methods, family values, family breakup, ineffective management and problem-solving in the family, and weak social capital in the family. The category of Ineffective interactions and social support also encompass two sub-categories: Peer pressure and reference groups, and inappropriate care and support relationship between teachers and students. More details were shown in Table 2.

## Individual factors

**1. Biological, psychological, and demographic factors.**   *1.1. Insufficient cognitive and inferential development.* Most of the girls stated that after being courted, they did not think about the suitor for a long time and quickly decided on marriage. They admitted that they did not know enough about marriage, pregnancy, or individual rights. They were unable to understand the complexities of life, lacked the power to make decisions and negotiate, so they left the decision of marriage to their parents.

*Girl said, "If I had known that life was so difficult, I would not have gotten married when I was a child. I thought that life was all about love and affection. I understood what marriage is only after marriage".*

*(Code 1, 15 years old, village)*

In most cases, girls did not want to continue their education or preferred to pursue it after marriage. They did not have a specific life goal; their primary aim was to start a family.

*Informant said: "Students say why should we study, if we study, there is no job, we will end up as a bride. Well, it is better to get married now".*

*(Code 33, consultant, 29 years of experience)*

Before marriage, girls used to talk with their friends about marriage, wedding ceremonies, makeup, hair styling, and wedding dresses. It was very pleasant to think about living with someone who loved them passionately.

*Girl said: "I always thought I should marry my first love. When Ali came to propose, I don't know why I fell in love with him. I had a suitor before this, but Ali was strangely close to me, I didn't want to lose him".*

*(Code 29, 9 years old, city)*

The need for affection and the desire for companionship were also significant driving forces in deciding to get married.

*Girl said: "I wanted someone to like me and buy me a gift, so that we could be engaged, but when it came to getting married, I was very young, I had just turned 13".*

*(Code 12, 20 years old, city)*

*1.2. Physiological and anatomical features.* The participants stated that being tall, having a large body, a feminine physique, and menstruating lead families and society to believe that

**Table 2. Categories, subcategories and codes obtained from the interviews.**

| Category | Subcategory | Code | Quotation |
|---|---|---|---|
| *Individual factors* | | | |
| **1. Biological, psychological, and demographic factors** | 1.1. Insufficient cognitive and inferential development | Instant decision-making, aimlessness in life, childish thoughts of love and infatuation, responding to the need, misunderstanding of life together, ineffectiveness of education, disappointment in education, lack of knowledge about marriage and pregnancy, lack of knowledge about individual rights, weakness of life skills. | I already loved her, when she came to propose, it was something like love, then my eyes and ears were blinded. (Girl) |
| | 1.2. Physiological and anatomical features | Tall, big body, monthly period, beauty, sex appeal | Although my daughter was 14 years old, she was tall and had a big body, but she was not petite like these children. (Father) |
| | 1.3. Facing stressful factors in life | Tension in the family, difficult living conditions, insecurity in the family, limited independence and freedom, experiencing failure, feeling alone | I am the only girl, I was very lonely, I wanted to get married. (Girl) |
| | 1.4. Demographic Characteristics | Baloch ethnicity, Sunni religion, living in the village | This type of marriage usually happens in rural areas. (Female informant) |
| *Interpersonal factors* | | | |
| **2. Family structure** | 2.1. Traditional Parenting methods | Restrictions on girls' social relationships, strengthening traditional roles, making girls responsible for the difficulty of growing up, encouraging and supporting the family, making decisions by parents, accepting suitors at a young age, accepting part of the responsibility of couples. | You teach them how to cook eggs since they were children in the second grade, teach them that their life will be full in 15 years. (Mother) |
| | 2.2. Family values | Possessing a social, economic and cultural status, having similar social status and being of the same class, having sufficient knowledge of girls and boys, being chaste, being a religious girl, being a responsible housewife, and being committed to high social relations, being modest. | I knew her well, she was a girl with a personality, she was not a girl who was naughty like the girls in the street and so on. (Husband) |
| | 2.3. Family breakup | Unrestrained family members, divorce, imprisonment, violence, addiction, adoption, bad luck with the daughter, arguments in the family, death of parents, physical and mental disability in the family. | If their parents are in prison, their children will get married sooner. In our school, we have students whose parents are in prison and who live with others and are forced to get married. (Female informant) |
| | 2.4. Inefficiency of management and problem solving in the family | Escaping from responsibility, not paying attention to education, providing financial resources by the mother, power and influence of the mother, disagreement between parents, the desire to expand the family, fear and worry of the family, loneliness of the mother, prevention of family disputes, acceptance of the son-in-law as a son. | Many suitors came every day. I married her so that they would not disturb her again. (Father) |
| | 2.5. Weak social capital in the family | Weak social relations of the family, low education of parents, low attachment among family members, ignorance of parents, absence of one parent at home, false job, single parent, only daughter, not caring about the child, lack of trust in the family. | These are usually families with uneducated parents. An educated family allows the girl to study, get a job, and marry someone at her own level. (Male informant) |
| **3. Ineffective interactions and social support** | 3.1. Peer pressure and reference groups | Stimulation of emotions, recommendation of peers, competition with peers, support of religious leaders, approval of relatives and relatives, media advertising | I saw that our daughter-in-law got married when she was 13 years old, so I said that I, who am 14 years old, can get married. (Girl) |
| | 3.2. Inappropriate care and support relationship between teacher and student | Counselor inefficiency, lack of motivation of the teacher, embarrassment of the student, weakness in responsibility, insufficient time for collective counseling | I said several times that I would go to counseling, but I did not go because I was afraid that the children would ask me questions, I was ashamed to talk about these things. (Girl) |

girls are ready for marriage and starting a family. This belief stems from the religious viewpoint of the family. In religious teachings, physical maturity is presented as one of the prerequisites for girls' marriage.

*Father said: "In our religion, it is recommended that the girl should have her period in her husband's house. However, we married our daughter when her period started".*

*(Code 24, 57 years old, police force)*

*1.3. Facing stressful factors in life.* Participants indicated that if life conditions become challenging for girls, they choose early marriage as a way to escape from this situation.

*Informant said, "I have a student who doesn't come to school because he has to take care of his siblings. His mother goes to work, and he doesn't have a father. So, in this situation, the best option for her is marriage."*

*(Code 31, teacher, 17 years of service).*

One of the characteristics of adolescence is the desire to gain independence and freedom. Girls tend to keep up with the changes in society and make decisions about their clothing choices, wanting to dress according to their own preferences and be visible in the community. In some families, this form of self-expression through clothing may not be accepted. If girls perceive their family as a barrier to their independence and freedom, they may be pushed towards marriage as a means to break free from their current circumstances and attain autonomy.

*Informant said, "Children want to be modern, be like everyone else, dress well, go to the hairdresser, dye their hair, but if the family doesn't let them, they think about getting married so that they can be comfortable, so that no one will have a job for them anymore".*

*(Code33, teacher, 18 years of experience service)*

In cases where the girls were the only child or the only girl in the family, the feeling of loneliness was unpleasant for them and increased their desire to get married. They considered marriage the only way to alleviate loneliness and establish a lasting relationship.

*Girl said: "I was the only girl in the family, I was alone at home a lot. I tried to spend time with my friend, but this relationship never lasted. I decided to get married instead".*

*(Code 3, 15 years old, city)*

*1.4. Demographic characteristics.* Although the marriage of girls under the age of 15 is more common among the Baloch and Sunni communities, many Persian and Shiite girls also get married at the age of 15.

*Informant said: "Child marriage is more common in the Baloch people. They are used to this type of marriage and it is not a strange thing".*

*(Code 33, education employee, 12 years of service)*

People who reside in less privileged areas, such as rural regions, are more inclined to marry at a young age due to various reasons. In rural and nomadic areas, the primary occupations revolve around animal husbandry and agriculture, where women are actively involved

alongside their household chores. Therefore, CM serves as a means to provide labor. Traditional practices hold greater significance in rural areas, and deviating from these customs often invites criticism from the community. Moreover, villagers typically have larger families, which influences their choice to marry early.

*Mothers said: "We live in the village and we teach our children that they should get married at the age of 13. They should learn how we lived".*

*(Code, 20, 47 years old, housewife)*

## Interpersonal factors

**2. Family structure.** This category included the subcategory of traditional parenting methods, family values, Family breakup, inefficiency of management and problem solving in the family, weak social capital in the family.

*2.1. Traditional parenting methods.* In rural and less privileged areas, traditional parenting methods persist. Parents pass down parenting practices from generation to generation, teaching their children in the same manner. Girls are betrothed from a young age and are expected to be prepared for marriage. They are taught household tasks like cooking and housekeeping during childhood. There is a belief that girls should experience life's challenges and take on life responsibilities akin to their mothers. In this educational approach, the decision to marry is typically made by parents and elder family members, leading to most girls beginning their lives based on their parents' choices.

*Mothers said: "When Fazeh was in the second grade, I taught her how to cook eggs, so she should also know how to live".*

*(Code, 18, 44 years old, housewife)*

*2.2. Family values.* Family values can be interpreted in two parts. The prevailing values in the girl's family, which were criteria for selecting suitors, include possessing social, economic, and cultural status, similarity in social status, belonging to the same class, adequate knowledge of the boy. The prevailing values in the boy's family include similarity in social status, belonging to the same class, adequate knowledge of the girl, cleanliness, religiosity, homemaking skills, responsibility, commitment, strong social connections, and modesty, which encouraged them to marry a younger girl.

In most cases, parents referred to a "good suitor" as someone with a favorable social, economic, and cultural standing. Getting acquainted with the suitor was also a factor that contributed to the marriage of girls.

*Mother said: "It's hard to find good suitors now, you can't trust boys. A suitor who is like us, who is in our class, is a good candidate for marriage".*

*(Code 20, 47 years old, housewife)*

On the other hand, getting to know the girl well enough, ensuring that she is clean, modest, religious, capable of being a housewife, responsible, committed, and has high social relations were the most important reasons for husbands to marry young girls.

Wives believe that it is difficult to trust girls in the current era. When girls become part of the community, they tend to assimilate into it. As girls mature, their knowledge grows, and

establishing relationships with them becomes more challenging. They prefer to marry young girls who are familiar to them and whose purity and decency they are certain of.

**2.3. Family breakup.** Divorce, imprisonment of parents, and death of parents weaken the foundation of the family, and girls lose the support of the family. In the earthquake of 2012, a large number of children lost their parents and were placed under the care of a family member.

*Girl said: "I got married because I was motherless. My mother died in an earthquake, my father was an addict, my uncle raised us, I was still a child, I didn't want to, but I had no choice".*

*(Code 8, 27 years old, city)*

The mistreatment of girls and the physical and mental incapacity of parents and the feeling of inability to take care of girls put girls at the risk of early marriage.

*Mother said: "I have nervous problems, I take several nerve pills a day, I'm not feeling well at all, maybe I won't be alive tomorrow. I wanted my daughter to calm down so that I can feel at ease".*

*(Code 17, 52 years old, employee)*

**2.4. Inefficiency of management and problem solving in the family.** During life, there are issues that parents have to decide on. The skill of parents in managing problems and issues plays an important role in the events of children's lives and future. The participants pointed out various concerns such as the desire to expand the family (having grandchildren), fear and worry within the family, the mother's loneliness, and the prevention of family disputes. These issues indicate ineffectiveness of parents in solving problems.

For example, most of the parents expressed concern about their daughters. They agreed to marry off the girls to protect them from harm and to relinquish the responsibility of caring for them.

The fathers said: "His mother was very insistent that we marry her, she said that something might happen and dishonor would happen, and I agreed" (Code 16, 60 years old, driver)

**2.5. Weak social capital in the family.** Weak social capital within the family can contribute to early marriage among girls. In families with few children, the connections between family members are restricted, resulting in a limited family support network that may negatively impact social capital. Conversely, in families with a greater number of children, while the relationships between family members may be more extensive, neglecting children is more prevalent, potentially diminishing the bond among family members. Low parental education, parental ignorance, parental employment status, and parental absence in the family are additional factors that influence social capital and impact the marriage of girls.

*Mother said: If my daughter had been in a safe place while I was working, If my daughter were in a safe place while I was working, I would have opposed the marriage. We worked 24 hours a day and we had 48 hours off. The situation has become very terrible, it was very difficult for me".*

*(Code 17, 52 years old, employee)*

**3. Ineffective interactions and social support.** *3.1. Peer pressure and reference groups.* The most crucial interactions for teenagers occur within the family, school, and society, influencing their decisions about marriage. On one hand, teenage girls yearn for romantic relationships, marriage, and envisioning their future lives. At school, married peers share stories about the beauty and happiness of life together. On the other hand, In traditional societies, relatives eagerly anticipate girls' marriages. Religious leaders advocate for early marriage for girls, and media advertisements also significantly impact this process.

*Girl said: " I used to watch wedding movies and such in cyberspace. I wanted to get married".*

*(Code 7, 20 years old, city)*

*3.2. Inappropriate care and support relationship between teacher and student.* The participants pointed out issues such as the ineffectiveness of counselors, lack of motivation among teachers, students' embarrassment, and group counseling. These factors indicate inappropriate care and support relationships between teachers and students, which contribute to the prevalence of CM.

Despite the establishment of counseling centers in schools, the presence of counselors is limited due to a shortage of human resources. The quantitative evaluation system for counselors' performance results in excessive documentation, consuming most of their time. Consequently, counselors often remain unaware of students' status and conditions. Furthermore, the lack of expertise in counseling hampers their effectiveness.

*Girl said: "We had counselors in middle school and high school, but it didn't help at all, so I didn't go anymore".*

*(Code 7, 20 years old, city)*

## Discussion

CM is a significant public health issue with numerous consequences for individuals and society. This concern has prompted the international community to address it in the 2030 sustainable development goals and to endorse policies aimed at controlling and preventing CM. Identifying the root causes of CM as the initial step can pave the way for targeted interventions in its prevention and control. this study has revealed that individual and interpersonal factors can influence CM.

Inadequate cognitive and inferential development stem from two biological and cognitive conditions and a lack of knowledge about marriage and fertility. By examining Piaget's cognitive stages, we find that during puberty, children enter the final stage of cognitive development, which marks the onset of abstract thinking. This ability develops gradually, and children are not yet equipped to make decisions about significant life events like marriage at this stage [26]. The study of how teenagers think about marriage shows that children are immature in understanding the complex dimensions of life and do not have the ability to comprehend interpersonal relationships, emotional sensitivity, and social perspective [27]. In Naghizadeh's et al study, instant decision-making, aimlessness in life, and childish thoughts were introduced as important factors influencing early marriage [28]. Eftikharzadeh also attributes this intellectual immaturity to aimlessness in life [29]. Some of the girls wanted to get married to fulfill their needs that were not being met by their families. The most important need expressed by the participants was emotional. Javadian et al believes that during adolescence, if the family fails to

meet the financial, emotional, recreational, and sexual needs of girls, it can lead them to marry at a young age [30]. Some participants viewed love for their spouse as a reason for marriage. Several studies have reported that at times, girls marry out of love for their partner, overlooking other considerations, which may indicate their lack of intellectual maturity [31–35]. In addition to children's biological conditions, girls' lack of knowledge about marriage, pregnancy, life responsibilities, and life skills has been proposed as one of the factors contributing to CM, a finding that has been confirmed in other studies [19, 36, 37].

In some societies, CM is justified under the pretext of initiating menstruation, gaining reproductive power, and the ability to become a mother. Evidence also supports this assertion [34, 38–40]. Rahimi et al argues that religious and cultural norms significantly influence the development of this belief [31]. The beauty and sexual attractiveness of girls can heighten men's desire for them. In Rahimi's et al study, besides facial and bodily beauty, the tone of voice and dialect of girls also amplified men's desire for young girls [31]. This inclination can instill a sense of apprehension in families that their daughters may not find suitable partners as they grow older [31]. The results showed that the presence of stressful factors in life, such as independence and limited freedom, feelings of loneliness can be a trigger for girls to marry. In several studies in Iran [31, 41], America [42], and Israel [43], restrictions on traveling and being accountable to parents were introduced as one of the reasons for marrying at a young age. Girls' desire to choose the type of clothing and adornment and freedom from family orders was another factor for CM. What Rahimi et al refers to as "becoming a woman". In other words, the possibility of doing feminine things such as applying make-up, wearing open and formal clothes, which are unacceptable for the family [31]. The evidence shows that the difference in socio-economic development of urban and rural areas is influenced by factors such as education [44], access to the Internet, media, and information [45, 46], empowerment of girls and women [47], job opportunities [48], the formation of traditional norms and beliefs [49], and vulnerability to natural disasters [50], which provide the context for CM.

Since marriage takes place within the family, the structure of the family plays a decisive role in the type and timing of marriage. It appears that most participants in the study resided in traditional families, and this structure influenced CM in various ways. According to McMaster's model, crucial aspects of family members' functioning include problem-solving, communication, roles, emotional responsiveness, emotional expression, and behavior control [51]. This is overshadowed by the traditional nature of the family. Some experts argue that families with a traditional structure exhibit poor performance [52] and encounter challenges in emotional matters, often struggling with the problem-solving process. In these families, communication is neglected, role assignments and responses are unclear [53]. Although these opinions cannot be definitively accepted, their role cannot be denied either.

In such a structure, girls' social relations are limited, and girls are prepared from childhood to play traditional roles in the family, such as cooking and taking care of children. It seems that the continuation of the traditional function in the family originates from the perceived place of women and girls in the culture of societies. Still, in areas like Iran [41], Bangladesh [54], Sudan [55], and Uganda [39], social, economic, and livelihood issues are based on the traditional "mechanical society. In this context, male characters are considered the engine of economic progress, livelihood, and security, while female roles are limited to domestic work and childbirth to ensure the continuation of the family generation.

Traditional beliefs and the fear of girls getting involved in romantic relationships lead parents to consent to marrying girls at a young age in order to attain peace of mind and alleviate worries. They hold the belief that post-marriage, they are no longer accountable for their daughters. This notion has been supported by other studies as well [41, 54, 56]. These convictions are so entrenched that they can impede girls' education. In reality, excluding girls from

school is a tactic to uphold cultural values and family honor, with priority given to this over the ability to afford education [41, 57–59]. This issue can be viewed as a reflection of the family's ineffectiveness in managing and resolving the problem. This operational method may be linked to parents' limited education and awareness [60, 61]. Participants mentioned that parents' occupations and the time spent at home can also influence the decision to marry off girls. Given that parents in rural areas are primarily engaged in agriculture and animal husbandry, they naturally have less time at home. Other studies have indicated that girls whose parents are farmers are more prone to early marriage. Farmers typically have lower education and socioeconomic status, which can contribute to early marriage among girls [44, 62]. In addition to this, factors such as divorce, addiction, and having imprisoned parents, which weaken and sometimes collapse the structure and cohesion of the family, expose children to early marriage. Studies also support this finding [29, 41].

After primary relationships with parents, students need teachers' support in secondary relationships. These relationships play a decisive role in children's psychological well-being and have become one of the most important and effective areas of social support. Therefore, it is important for teachers to be careful in their relationships with students and to understand their issues and problems [63, 64]. However, the education system has not made much progress in responding to this need of students [65]. Inefficiency of school counselors, lack of motivation of teachers, weakness in their responsibility diminish the protective role of the school against the damage to students. In Momeni's et al study, the students expressed characteristics such as ability in human relations, appropriate personality traits, proper guidance and counseling, job commitment, spending time with the student as the characteristics of a supportive teacher. Failure to form supportive relationships can cause disinterest in studies and provide grounds for dropping out of school [66].

Modern approaches to adolescence emphasize that the positive and negative development of adolescents is influenced by multifaceted relationships. These relationships include the individual's biological and psychological characteristics on one hand, and family, culture, physical, and ecological environment on the other hand. After the family, which serves as the primary source of education and care, teenagers strive to establish relationships beyond the family, particularly among friends. In many cases, peers and friends act as guides for social behavior encourage girls to marry, the evidence also supports this finding [32, 34, 67], in addition to peers, relatives [32, 42, 68]. Media advertisements [41] and leaders' speeches [69] are effective in persuading CM.

Young girls who marry early often have a limited communication network and lack a strong support system. They receive insufficient support from their families, schools, and communities, and peer support is also lacking. Given the significant amount of time students spend in school, the role of educational institutions in fostering a communication and support network should not be overlooked. Teachers play a crucial role by establishing close and friendly relationships with their students, which enables them to positively influence students' decisions and behaviors. Furthermore, schools serve as a bridge between students and their families, helping to identify and address communication gaps. It is essential to empower parents and encourage family participation in finding appropriate solutions, including functions that schools can effectively manage.

The girls' support network extends beyond the school and is strengthened by engaging the local community, launching local campaigns [70], and involving religious leaders [71]. These leaders play a crucial role in transforming cultural and social norms. In addition to influencing these norms and advocating for girls, popular campaigns, in collaboration with religious leaders, can encourage government involvement and secure additional funding to address the needs of girls and support their empowerment. Engaging community members is essential for

identifying effective solutions that benefit the local population. Furthermore, the involvement of men and boys, who hold significant social power in traditional cultures, should not be overlooked [72]. Government agencies and non-governmental organizations can also contribute to this collective effort by supporting orphaned and single-parent children, expanding educational opportunities, and promoting local initiative.It is evident that the implementation of these programs necessitates national policies and a revision of existing laws. For instance, in Iran, Article 1041 of the Civil Code permits the marriage of a girl before she reaches the age of 13 and a boy before he turns 15, provided that they obtain permission from a guardian and that the marriage is deemed expedient by a competent court. The legality of child marriage under the age of 15 encourages families to marry off their children. There are two critical issues within this law that require revision. First, the age of marriage is not clearly defined; according to the law regarding maturity, girls under the age of 13 can also be married. Second, the requirement for parental permission and the court's determination of expediency to issue a marriage license raises concerns. In 30 countries around the world, a significant flaw exists in the legal framework regarding the marriage of girls under the age of 18, which is often arranged with parental consent [73]. This issue undermines the deterrent effect of the law. For instance, current legislation perpetuates a cultural belief that marriage serves as a means to protect girls from premarital sex [74]. Furthermore, when parents arrange early marriages for financial gain, it effectively legitimizes a harmful practice [75]. Therefore, establishing a legal age limit for marriage and revising existing laws could be effective solutions to reduce child marriage. Additionally, expanding research to investigate various aspects of child marriage and promoting evidence-based decision-making can facilitate the implementation of targeted interventions [76].

## Strengths and limitations

For the first time, this research identified individual and interpersonal factors affecting child marriage in Bam, Iran. Also, this study is one of the few studies that investigated the factors contributing to child marriage in girls under 15 years old. However, there were limitations in this research. We were unable to include a diverse sample in terms of Sunni religion. Additionally, the absence of other informants, such as religious leaders and healthcare workers, was another weakness of the study.

## Conclusion

The results indicate that physiological, psychological, and demographic characteristics are the most significant factors contributing to child marriage at the individual level. At the interpersonal level, family structure, ineffective interactions, and a lack of social support are the primary influences. Given that child marriage is a complex issue shaped by various factors, it is essential to develop a tailored set of strategies to address it, taking into account the cultural and contextual conditions of each society. Changes in legal frameworks can play a crucial role in preventing early marriages. Evidence suggests that in countries where the minimum legal age for marriage is below 18, nearly 100 million girls remain unprotected by national laws against child marriage [73]. Changes in laws, combined with educational campaigns, can raise awareness about the negative effects of child marriage [77]. Additionally, empowering local communities through robust support networks can help identify at-risk girls and provide the necessary resources to prevent early marriages. By engaging community leaders and utilizing social media platforms, these networks can foster a more informed and active community [78].

## Supporting information

**S1 Table. Standards for Reporting Qualitative Research (SRQR).**
(DOCX)

**S2 Table. Demographic characteristics of women's who married under the age of 15.**
(DOCX)

## Acknowledgments

The authors thank all the participants who helped us find these results.

## Author Contributions

**Conceptualization:** Asma pourtaheri, Mehr Sadat Mahdizadeh, Hadi Tehrani, Nooshin Peyman.

**Data curation:** Asma pourtaheri, Mehr Sadat Mahdizadeh, Nooshin Peyman.

**Formal analysis:** Asma pourtaheri, Mehr Sadat Mahdizadeh, Hadi Tehrani, Nooshin Peyman.

**Funding acquisition:** Asma pourtaheri.

**Investigation:** Asma pourtaheri, Mehr Sadat Mahdizadeh, Nooshin Peyman.

**Methodology:** Asma pourtaheri, Mehr Sadat Mahdizadeh, Hadi Tehrani, Nooshin Peyman.

**Project administration:** Mehr Sadat Mahdizadeh, Jamshid Jamali, Nooshin Peyman.

**Resources:** Mehr Sadat Mahdizadeh, Nooshin Peyman.

**Software:** Asma pourtaheri, Mehr Sadat Mahdizadeh.

**Supervision:** Mehr Sadat Mahdizadeh, Hadi Tehrani, Nooshin Peyman.

**Validation:** Asma pourtaheri, Mehr Sadat Mahdizadeh, Nooshin Peyman.

**Visualization:** Asma pourtaheri, Mehr Sadat Mahdizadeh, Nooshin Peyman.

**Writing – original draft:** Asma pourtaheri, Mehr Sadat Mahdizadeh, Hadi Tehrani, Jamshid Jamali, Nooshin Peyman.

**Writing – review & editing:** Asma pourtaheri, Mehr Sadat Mahdizadeh, Hadi Tehrani, Jamshid Jamali, Nooshin Peyman.

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
