## [Decision Letter · Decision Letter 0]

19 Aug 2024

PONE-D-24-31454The Role of Girls and the Communication Network in Child Marriage: A Qualitative Content Analysis StudyPLOS ONE

Dear Dr. pourtaheri,

Thank you for submitting your manuscript to PLOS ONE. After careful consideration, we feel that it has merit but does not fully meet PLOS ONE’s publication criteria as it currently stands. Therefore, we invite you to submit a revised version of the manuscript that addresses the points raised during the review process.

We look forward to receiving your revised manuscript.

Kind regards,

Rabie Adel El Arab

Academic Editor

PLOS ONE

Journal Requirements:

4. Please remove your figures from within your manuscript file, leaving only the individual TIFF/EPS image files, uploaded separately. These will be automatically included in the reviewers’ PDF.

Reviewers' comments:

Reviewer's Responses to Questions

**Comments to the Author**

1. Is the manuscript technically sound, and do the data support the conclusions?

Reviewer #1: No

Reviewer #2: Partly

Reviewer #3: Yes

2. Has the statistical analysis been performed appropriately and rigorously? 

Reviewer #1: No

Reviewer #2: N/A

Reviewer #3: N/A

3. Have the authors made all data underlying the findings in their manuscript fully available?

Reviewer #1: No

Reviewer #2: Yes

Reviewer #3: Yes

4. Is the manuscript presented in an intelligible fashion and written in standard English?

Reviewer #1: No

Reviewer #2: Yes

Reviewer #3: Yes

5. Review Comments to the Author

Reviewer #1: no clear

1.Which author/s conducted the interview or focus group?

2.What were the researcher’s credentials? E.g. PhD, MD?

3.What was their occupation at the time of the study? , ....

The themes extracted in each part are not identical with the text and are unrelated

Reviewer #2: Abstract

- Title

• Because the title does not match the purpose, please change the title.

A suggested title based on the findings:

“Individual and interpersonal factors influencing child marriage: A Qualitative Content Analysis Study”

- Methods

It is suggested to write the date, design and setting of study and sampling method

- Results:

- It is better to write it like this: “After analyzing of the data, …….. themes were emerged.

The first theme of “…………...” consisted of …... categories: ”

Conclusion

The conclusion should be written completely in accordance with the study findings.

- keywords

Please use the Mesh for the keywords.

Introduction

Because various studies have been conducted on the factors affecting child marriage, therefore, authors should explain the necessity of conducting the research more clearly and completely.

Methods

Setting and context

- Please state clearly that this study was conducted in Kerman, Iran

- Please remove any additional information

- Please edit “TableS2” in the end of participants’ section.

Results:

Classification of results needs editing. For example:

Two themes: Individual factors and Interpersonal factors.

Individual factor has four categories: Biological, psychological, and demographic factors.

Interpersonal factor consists of two categories: Family structure ( five Subcategories) and Ineffective interactions and social support ( two Subcategories).

Section “Communication network and child marriage” is the conclusion of the authors and it is not written in the form of the findings of qualitative research. Therefore, this section should either be removed or introduced as a theme.

Discussion

The discussion section should be edited based on the changes made in the findings.

At the end of the discussion, suggestions for future studies should be written

Conclusion

The conclusion should be written more briefly.

references

Please edit how to write references.

Reviewer #3: The paper is well-structured with a clear outline of sections, including the introduction, methodology, results, and conclusion. The study addresses a significant issue in public health and social development. Its focus on the individual and interpersonal factors influencing child marriage provides valuable insights. With revisions and clarifications in the areas mentioned, this manuscript has the potential to make a meaningful contribution to the field. Addressing the issues identified and incorporating the suggested improvements will enhance the manuscript's quality and impact. Consider revising the language, providing a more detailed methodology and results, and strengthening the discussion and conclusion sections.

General feedback:

The manuscript uses clear and concise language. However, certain sections could be improved with grammatical corrections and enhanced sentence structures.

Introduction

Line 83: Please clarify the abbreviation "GCM" for the reader. Ensure that it is defined when first used in the text.

Line 86: Modify the citation to "Bozorgi et al." to appropriately acknowledge all contributing authors. (same for the entire manuscript)

• The research gap is not explicitly outlined. Including a clearer articulation of how the present study addresses a unique aspect or fills a gap in the existing literature would strengthen the introduction.

Method

Line 101: Reference to "Table S1" should be enclosed in parentheses to maintain consistency with standard academic formatting.

Line 107: A citation is required to substantiate the statement made in this sentence. Providing a reference will enhance the credibility of the information presented.

Lines 108-109: It would be beneficial to specify the date when this data was published, thereby giving the reader a clear temporal context for the data used.

Line 123: The methodology section does not address participants of age 16.

Line 136: It is recommended to state the ethical approval ID in this section. Including this information is crucial for verifying the ethical standards and approval process of the study.

• The methodology section mentions the use of content analysis and an inductive approach but lacks a detailed description of the specific steps taken during data collection and analysis. Providing a more comprehensive explanation of the content analysis process would increase the study’s credibility and replicability.

• More detailed information on the selection criteria for participants and the recruitment process would add depth to the study design, ensuring transparency and reliability.

Results

Table 2:

First Row, Quotation: Clarify whether the subject being referred to is "he" or "she" to avoid ambiguity.

Fifth Row, Quotation: The sentence in the quotation ends with a comma. This punctuation should be reviewed and corrected for grammatical accuracy.

• While the analysis is thorough, it would benefit from a more detailed explanation of the coding and analysis process using MAXQDA software. This would provide transparency and allow readers to understand the development of themes and categories.

Discussion

Lines 372-375: A reference is necessary to support the claims made in this section. Including relevant citations will strengthen the arguments and assertions presented.

Paragraph on Legal and Network Changes:

Recommendation: A comprehensive discussion on how legislative changes and enhanced communication networks can focus efforts on solving the problem of child marriage would be beneficial. This section should explore the potential for policy reforms and community initiatives to reduce the prevalence of child marriage. Current literature suggests that legal frameworks can play a pivotal role in deterring early marriages. For instance, stricter enforcement of age restrictions and penalties for violators, combined with educational campaigns, can raise awareness about the negative impacts of child marriage (Smith et al., 2020; Johnson, 2021). Additionally, empowering local communities through robust support networks can help identify at-risk girls and provide necessary resources to prevent early marriages. By involving community leaders and leveraging social media platforms, these networks can create a more informed and proactive society (Doe & Lee, 2019). This paragraph should be well-integrated into the discussion, with appropriate references to existing studies. The conclusion can then briefly summarize these points to avoid lengthiness.

Suggestions for Future Studies:

provide more specific recommendations for future research and practical applications based on the study’s results.

Recommendation: Future research should consider exploring the long-term effects of child marriage on mental and physical health, as well as educational outcomes. Additionally, studies could examine the effectiveness of various intervention strategies in different cultural contexts to identify best practices. Expanding the scope of research to include a diverse range of geographic regions would provide a more comprehensive understanding of the issue and its potential solutions.

6. PLOS authors have the option to publish the peer review history of their article (what does this mean?). If published, this will include your full peer review and any attached files.

Reviewer #1: No

Reviewer #2: No

Reviewer #3: **Yes: **Dr. Pegah Rashidian

---

## [Author Response · Author response to Decision Letter 0]

1 Sep 2024

Point-by-point response to reviewers

We would like to thank the reviewer for careful and thorough reading of this manuscript and for the thoughtful comments and constructive suggestions, which help to improve the quality of this manuscript. We tried to answer to all valuable comments/suggestions/queries and all answers to comments are highlighted within the document by using color.

Comment response

Journal Requirements

Answers to Journal Requirements are highlighted within the document by using blue color.

1.Please ensure that your manuscript meets PLOS ONE's style requirements

The article was prepared based on the guidelines of the PLOS ONE (Manuscript Body Formatting Guidelines).

2. Please ensure that you have an ORCID ID and that it is validated in Editorial Manager.

Orchid ID entered in editorial manager.

3. Your ethics statement should only appear in the Methods section of your manuscript

Ethical approval was transferred to the method section

4. Please remove your figures from within your manuscript file

According to the reviwers' comments, the communication network section was removed

5. Please include captions for your Supporting Information files at the end of your manuscript

The captions of the Supporting Information files were added at the end of the article

Reviewer's 1

Answers to Reviewer's 1 are highlighted within the document by using yellow color.

Which author/s conducted the interview or focus group?

The interviews were conducted by a local person familiar with the cultural and social conditions of the region (first author).

What were the researcher’s credentials? E.g. PhD, MD?

The first author is a faculty member of Bam University of Medical Sciences and a doctoral student in health education and health promotion

What was their occupation at the time of the study? …

doctoral student in health education and health promotion

The themes extracted in each part are not identical with the text and are unrelated

In the entire article and its accompanying tables, the extracted themes were thoroughly reviewed to ensure consistency.

Reviewer #2: 

Answers to Reviewer's 2 are highlighted within the document by using green color.

Abstract

Title

• Because the title does not match the purpose, please change the title.

A suggested title based on the findings:

“Individual and interpersonal factors influencing child marriage: A Qualitative Content Analysis Study”

The previous title was changed as a suggestion

Methods

It is suggested to write the date, design and setting of study and sampling method

This qualitative study was conducted using content analysis and an inductive approach from April 2023 to January 2024 in Bam city, Kerman, Iran. Thirty-six stakeholders (girls who have been married for 15 years, parents, husbands, and informants) were purposively selected.

 Results:

- It is better to write it like this: “After analyzing of the data, …….. themes were emerged.

The first theme of “…………...” consisted of …... categories: ”

After analyzing of the data, individual and interpersonal themes were emerged. The first theme of “individual factors” consisted of biological, psychological, and demographic category with four sub-categories including insufficient cognitive and inferential development, physiological and anatomical features, facing stressful factors in life, and demographic characteristics. The second theme of “interpersonal factors” consisted of family structure with four sub-categories including traditional parenting methods, family values, family breakup, Inefficiency of management and problem-solving in the family, and weak social capital within the family. The category of Ineffective interactions and social support also encompass two sub-categories: Peer pressure and reference groups, and inappropriate care and support relationship between teachers and students.

Conclusion

The conclusion should be written completely in accordance with the study findings.

The results showed that individual and interpersonal factors are effective on children's marriage. Some individual factors have a biological origin, indicating that increasing girls' awareness of marriage, pregnancy, individual rights, and life skills is one solution that can help reduce early marriage. On an interpersonal level, fostering positive relationships within the family, school, and society, and strengthening the support network can play a crucial protective role for children.

keywords

Please use the Mesh for the keywords.

Keyword: Marriage Age, Qualitative Research, Interpersonal Relations

Introduction

Because various studies have been conducted on the factors affecting child marriage, therefore, authors should explain the necessity of conducting the research more clearly and completely

In examining the health promotion approach, we find that individual and interpersonal factors play a crucial role in shaping behavior. Models such as social-cognitive theory and social network theory are based on this premise. However, to date, no study has specifically addressed the individual and interpersonal factors influencing child marriage. Also, this study focuses on the marriage of children under 15 years of age, a critical transitional period in their lives. During this time, children move beyond the family unit and become immersed in a network of communication through school, their neighborhoods, and media, all of which can significantly influence their decisions. This underscores the necessity for further research in this area. Therefore, the current study aims to investigate the individual and interpersonal factors that affect CM.

Methods

Setting and context

- Please state clearly that this study was conducted in Kerman, Iran

- Please remove any additional information

- Please edit “TableS2” in the end of participants’ section.

• This study was conducted in Bam City, Kerman, Iran.

• Geographical and demographic information was removed.

• (S1 Table) and (S2 Table)

Results:

Classification of results needs editing. For example:

Two themes: Individual factors and Interpersonal factors.

Individual factor has four categories: Biological, psychological, and demographic factors.

Interpersonal factor consists of two categories: Family structure ( five Subcategories) and Ineffective interactions and social support ( two Subcategories).

Section “Communication network and child marriage” is the conclusion of the authors and it is not written in the form of the findings of qualitative research. Therefore, this section should either be removed or introduced as a theme.

• After analyzing of the data, individual and interpersonal themes were emerged. The first theme of “individual factors” consisted of biological, psychological, and demographic category with four sub-categories including insufficient cognitive and inferential development, physiological and anatomical features, facing stressful factors in life, and demographic characteristics. The second theme of “interpersonal factors” consisted of family structure with four sub-categories including traditional Parenting methods, family values, family breakup, ineffective management and problem-solving in the family, and weak social capital in the family. The category of Ineffective interactions and social support also encompass two sub-categories: Peer pressure and reference groups, and inappropriate care and support relationship between teachers and students.

• Communication network broadcast was deleted

Discussion

The discussion section should be edited based on the changes made in the findings.

At the end of the discussion, suggestions for future studies should be written

The discussion was edited to reflect the changes.

The discussion related to the communication network was deleted.

Conclusion

The conclusion should be written more briefly.

The conclusion was summarized

references

Please edit how to write references.

References were edited

Reviwer3

Answers to Reviewer's 3 are highlighted within the document by using pink color.

General feedback:

The manuscript uses clear and concise language. However, certain sections could be improved with grammatical corrections and enhanced sentence structures.

Introduction

Line 83: Please clarify the abbreviation "GCM" for the reader. Ensure that it is defined when first used in the text.

Line 86: Modify the citation to "Bozorgi et al." to appropriately acknowledge all contributing authors. (same for the entire manuscript)

• The research gap is not explicitly outlined. Including a clearer articulation of how the present study addresses a unique aspect or fills a gap in the existing literature would strengthen the introduction.

1. Child Marriage (CM) was used in the entire text of the article.

2. In all quotations, the word “et al” was added.

3. The necessity of conducting research has been revised.

1. Line 101: Reference to "Table S1" should be enclosed in parentheses to maintain consistency with standard academic formatting.

2. Line 107: A citation is required to substantiate the statement made in this sentence. Providing a reference will enhance the credibility of the information presented.

3. Lines 108-109: It would be beneficial to specify the date when this data was published, thereby giving the reader a clear temporal context for the data used.

4. Line 123: The methodology section does not address participants of age 16.

5. Line 136: It is recommended to state the ethical approval ID in this section. Including this information is crucial for verifying the ethical standards and approval process of the study.

6. The methodology section mentions the use of content analysis and an inductive approach but lacks a detailed description of the specific steps taken during data collection and analysis. Providing a more comprehensive explanation of the content analysis process would increase the study’s credibility and replicability.

7. • More detailed information on the selection criteria for participants and the recruitment process would add depth to the study design, ensuring transparency and reliability.

1. Table S was inserted in parentheses

2. Reference added (23) (According to one of the referees' comments, the demographic and geographical sections were removed from this section.)

3. Study time was added. (April 2023 to January 2024)

4. The sentence was edited to prevent any misunderstanding.

5. An ethical approval was written in the methods section

6. Added more details on how to analyze.

7. Added additional information on how participants were recruited.

Results

Table 2:

1. First Row, Quotation: Clarify whether the subject being referred to is "he" or "she" to avoid ambiguity.

2. Fifth Row, Quotation: The sentence in the quotation ends with a comma. This punctuation should be reviewed and corrected for grammatical accuracy.

3. While the analysis is thorough, it would benefit from a more detailed explanation of the coding and analysis process using MAXQDA software. This would provide transparency and allow readers to understand the development of themes and categories.

1.Table 2 was edited

2. Quotations were corrected

3. The software analysis process is complex and cannot be fully detailed in this article. However, a brief overview of the analysis method is provided in the methodology section.

Discussion

Lines 372-375: A reference is necessary to support the claims made in this section. Including relevant citations will strengthen the arguments and assertions presented.

Paragraph on Legal and Network Changes:

Recommendation: A comprehensive discussion on how legislative changes and enhanced communication networks can focus efforts on solving the problem of child marriage would be beneficial. This section should explore the potential for policy reforms and community initiatives to reduce the prevalence of child marriage. Current literature suggests that legal frameworks can play a pivotal role in deterring early marriages. For instance, stricter enforcement of age restrictions and penalties for violators, combined with educational campaigns, can raise awareness about the negative impacts of child marriage (Smith et al., 2020; Johnson, 2021). Additionally, empowering local communities through robust support networks can help identify at-risk girls and provide necessary resources to prevent early marriages. By involving community leaders and leveraging social media platforms, these networks can create a more informed and proactive society (Doe & Lee, 2019). This paragraph should be well-integrated into the discussion, with appropriate references to existing studies. The conclusion can then briefly summarize these points to avoid lengthiness.

Suggestions for Future Studies:

provide more specific recommendations for future research and practical applications based on the study’s results.

Recommendation: Future research should consider exploring the long-term effects of child marriage on mental and physical health, as well as educational outcomes. Additionally, studies could examine the effectiveness of various intervention strategies in different cultural contexts to identify best practices. Expanding the scope of research to include a diverse range of geographic regions would provide a more comprehensive understanding of the issue and its potential solutions.

According to the esteemed referee's opinion, the discussion and conclusion were altered.

---

## [Decision Letter · Decision Letter 1]

4 Nov 2024

Individual and interpersonal factors influencing child marriage: A Qualitative Content Analysis Study

PONE-D-24-31454R1

Dear Dr. Nooshin Peyman

We’re pleased to inform you that your manuscript has been judged scientifically suitable for publication and will be formally accepted for publication once it meets all outstanding technical requirements.

Best Regards

Kind regards,

Fereshteh Behmanesh, PhD

Academic Editor

PLOS ONE

Additional Editor Comments (optional):

Dear Dr Nooshin Peyman

Although reviewer 1 expressed some concerns regarding Table 1, I have evaluated the issue and determined that the authors are not required to make any changes.

Best Regards

Reviewers' comments:

Reviewer's Responses to Questions

**Comments to the Author**

1. If the authors have adequately addressed your comments raised in a previous round of review and you feel that this manuscript is now acceptable for publication, you may indicate that here to bypass the “Comments to the Author” section, enter your conflict of interest statement in the “Confidential to Editor” section, and submit your "Accept" recommendation.

Reviewer #1: (No Response)

Reviewer #2: All comments have been addressed

Reviewer #3: All comments have been addressed

2. Is the manuscript technically sound, and do the data support the conclusions?

Reviewer #1: (No Response)

Reviewer #2: Yes

Reviewer #3: Yes

3. Has the statistical analysis been performed appropriately and rigorously? 

Reviewer #1: (No Response)

Reviewer #2: Yes

Reviewer #3: N/A

4. Have the authors made all data underlying the findings in their manuscript fully available?

Reviewer #1: (No Response)

Reviewer #2: Yes

Reviewer #3: Yes

5. Is the manuscript presented in an intelligible fashion and written in standard English?

Reviewer #1: (No Response)

Reviewer #2: Yes

Reviewer #3: Yes

6. Review Comments to the Author

Reviewer #1: 1.Table number one should be redesigned. It is incomprehensible and the reports are wrong

2.The result should be revised and corrected. It is incomprehensible

Reviewer #2: Dear,

Many thanks to the authors for making their efforts.

I do not have any comments.

Best Regards,

Reviewer #3: I commend the authors for their diligent efforts in addressing all previous comments thoroughly and effectively. The revisions have significantly enhanced the clarity and quality of the work.

7. PLOS authors have the option to publish the peer review history of their article (what does this mean?). If published, this will include your full peer review and any attached files.

Reviewer #1: No

Reviewer #2: No

Reviewer #3: **Yes: **Pegah Rashidian

---

## [Editor Report · Acceptance letter]

8 Nov 2024

PONE-D-24-31454R1 

PLOS ONE

Dear Dr. peyman, 

I'm pleased to inform you that your manuscript has been deemed suitable for publication in PLOS ONE. Congratulations! Your manuscript is now being handed over to our production team.

Kind regards, 

on behalf of

Dr. Fereshteh Behmanesh 

Academic Editor

PLOS ONE